# Development of a Novel In Vitro Model to Study Lymphatic Uptake of Drugs via Artificial Chylomicrons

**DOI:** 10.3390/pharmaceutics15112532

**Published:** 2023-10-26

**Authors:** Malaz Yousef, Chulhun Park, Mirla Henostroza, Nadia Bou Chacra, Neal M. Davies, Raimar Löbenberg

**Affiliations:** 1Faculty of Pharmacy and Pharmaceutical Sciences, University of Alberta, Edmonton, AB T6G 2E1, Canada; malaz@ualberta.ca (M.Y.); raimar@ualberta.ca (R.L.); 2College of Pharmacy, Jeju National University, Jeju 63243, Republic of Korea; chpark@jejunu.ac.kr; 3Faculty of Pharmaceutical Sciences, University of Sao Paulo, Sao Paulo 05508-000, Brazil; mirlabazan@usp.br (M.H.); chacra@usp.br (N.B.C.)

**Keywords:** model, lymphatic uptake, drug delivery, chylomicrons, drug absorption

## Abstract

The lymphatic system plays a crucial role in the absorption of lipophilic drugs, making it an important route for drug delivery. In this study, an in vitro model using Intralipid^®^ was developed to investigate the lymphatic uptake of drugs. The model was validated using cannabidiol, halofantrine, quercetin, and rifampicin. Remarkably, the uptake of these drugs closely mirrored what would transpire in vivo. Furthermore, adding peanut oil to the model system significantly increased the lymphatic uptake of rifampicin, consistent with meals containing fat stimulating lymphatic drug uptake. Conversely, the inclusion of pluronic L-81 was observed to inhibit the lymphatic uptake of rifampicin in the model. This in vitro model emerges as a valuable tool for investigating and predicting drug uptake via the lymphatic system. It marks the first phase in developing a physiologically based predictive tool that can be refined further to enhance the precision of drug interaction predictions with chylomicrons and their subsequent transport via the lymphatic system. Moreover, it can be employed to explore innovative drug formulations and excipients that either enhance or hinder lymphatic drug uptake. The insights gained from this study have significant implications for advancing drug delivery through the lymphatic system.

## 1. Introduction

Most orally administered drugs enter the systemic circulation after absorption from enterocytes in the intestine via the portal vein [1]. However, some orally administered lipophilic xenobiotics access the systemic vasculature via the intestinal lymphatic system, offering several advantages for drug delivery [2,3,4]. Absorption through the intestinal lymphatics enables the drug to reach the general circulation without passing through the liver, which is beneficial for molecules that are prone to first-pass metabolism by the liver [5,6]. Additionally, drug delivery through the lymphatic system can result in higher drug concentrations in the systemic circulation and improve bioavailability, as reported for several drugs [6,7,8]. The lymphatic route of absorption can be more effective for drugs targeting conditions involving the lymphatic system pathophysiology, including some viral infections, metabolic and inflammatory conditions, hypertension, and solid tumors, among others [5,9,10].

Drugs can access the intestinal lymphatics through enterocytes or microfold cells (M cells) in the follicle-associated epithelium (FAE) that overlies Peyer’s patches [1,2,6]. Chylomicrons, which are mainly composed of triglycerides, phospholipids, proteins, cholesterol, and cholesteryl esters, are responsible for transporting drugs across the enterocytes and into the lymphatic system [11,12]. Ingested triglycerides are hydrolyzed by lipases into monoacylglycerols and free fatty acids, which are subsequently re-esterified to triglycerides and assembled into chylomicrons in the endoplasmic reticulum and Golgi apparatus of enterocytes before being exocytosed via the basolateral membrane [1,2,3].

There are various methods of enhancing the transport of drugs through the lymphatic system. These include administration of drugs during a postprandial state and the utilization of lipid-based formulations and lipidic prodrugs [6]. Lipid-based nanoparticles have emerged as promising candidates for drug delivery. Multiple studies have been dedicated to creating formulations that leverage the potential of the intestinal lymphatics [1,2,6,12]. For instance, in a study involving Solid Lipid Nanoparticles (SLNs) loaded with nimodipine, a considerable improvement in bioavailability was observed compared to the drug solution. The bioavailability of nimodipine increased by a factor of 2.08 in male Albino Wistar rats. This enhancement was attributed to the portion of nimodipine carried by the lymphatic system [13].

Another illustration that highlights the harmonious interplay between utilizing a prodrug approach and the influence of food on the transportation of substances through the intestinal lymphatics can be found in the example of testosterone. When administered orally, testosterone exhibited limited effectiveness in treating male androgen deficiency syndromes due to significant initial loss during the first pass through the liver [14]. However, studies have indicated that testosterone undecanoate, a prodrug of testosterone, displays higher systemic exposure compared to testosterone administration [15,16]. Furthermore, the exposure to both testosterone and the prodrug was notably greater in individuals who had consumed a meal compared to those who were fasting. This increase in systemic exposure after a meal was correlated with an augmentation in lymphatic transport efficiency. Moreover, the strategic conjugation of testosterone with a lipophilic long-chain ester (undecanoate) led to an elevated androgenic response compared to using pure testosterone. This lipophilic ester was observed to follow the intestinal lymphatic route, evading the liver and ameliorating the initial pass effect. Once within the systemic circulation, the ester form underwent hydrolysis, liberating the free form of testosterone. Notably, a significant portion—ranging from 91.5% to 99.7%—of the testosterone available within the system was attributed to the testosterone undecanoate that had been transported via the lymphatic system [15]. Numerous additional instances of such approaches can be found within various reports and reviews in the existing literature [2,5,6,7].

For investigating lymphatic uptake, researchers have established and employed the mesenteric lymphatic cannulation model, which serves to assess lymphatic transport [17,18]. As intestinal lymph drains, it passes through the mesenteric lymph node en route to the circulation [5]. This model facilitates the direct sampling of the mesenteric lymph node, allowing for the quantification of the drug that enters the systemic circulation through its journey in the intestinal lymphatics. While this method simplifies sample collection and measurement, it is important to note that lymphatic cannulation is invasive and comes with inherent limitations. The procedure necessitates intricate surgical steps and may potentially influence lymph flow and vessel pressure gradients, rendering sequential sampling challenging following multiple sessions. Given the interplay of various factors, the overall success rate of lymphatic cannulation tends to be quite low [6].

To circumvent the challenges associated with cannulation procedures, researchers have introduced lymph blocking models by employing specific blocking agents. For instance, the use of cycloheximide, a protein synthesis inhibitor, offers a targeted approach to suppressing the exocytosis and secretion of chylomicrons from enterocytes [19]. Moreover, various other inhibitors employing distinct blocking mechanisms, including Pluronic L-81 and colchicine, have been utilized for assessing the portion of substances delivered through the intestinal lymph [20].

The in vivo models offer the best evaluation, yet they pose technical challenges, demanding advanced surgical expertise, while also raising ethical concerns regarding human application [6,10]. An alternative approach involves the utilization of in vitro models, incorporating both cellular and non-cellular strategies. Among cellular models, Caco-2 cells are commonly employed, but they come with constraints that include their resemblance to almost colonic rather than small intestinal tissue and the absence of the mucus layer secreted by goblet cells. Moreover, they lack other cell types existing in the human mucosa niche such as M cells [9]. Advancement has been achieved through the co-culture of Caco-2 cells with lymphoid cells like Raji cells, which exhibit M cell-mimicking capabilities, or murine-derived lymphocytes [21]. Nonetheless, this approach faces limitations due to the excessive presence of M-like cells, possibly leading to an overestimation of experimental outcomes. Consequently, there is a high anticipation for further enhancements or potential alternatives.

Regarding formulations targeted at lymphatic delivery and utilizing lipid-based carriers, the typical progression involves initial vehicle degradation via lipolysis. These degraded components subsequently transition into micellar vehicles, which are then absorbed by enterocytes and reassembled with chylomicrons [1,2]. To evaluate the effectiveness of such formulations, in vitro lipolysis has proven useful. This method estimates the proportion of the drug intended for intestinal lymphatic uptake by quantifying the fraction incorporated into the generated micellar vesicles after the completion of lipolysis [22].

The interaction between drugs and chylomicrons plays a pivotal role in facilitating drug transportation across enterocytes via lipoproteins carried by the lymphatic system. Notably, a direct correlation between drug uptake by isolated chylomicrons and in vivo intestinal lymphatic uptake has been documented across nine lymphotropic compounds. This finding highlights the credibility of using drug-chylomicron association as a reliable indicator for assessing intestinal lymphatic uptake [23,24].

The aim of this study is to develop a model with the use of artificial chylomicrons (Intralipid^®^) to investigate intestinal lymphatic uptake. Intralipid^®^ closely mimics the size and composition of natural chylomicrons, as illustrated in Figure 1 [11,25]. To examine the uptake of lymphotropic drugs known to undergo lymphatic absorption, a Franz cell diffusion system was employed. Cannabidiol, halofantrine, quercetin and rifampicin were used as model lymphotropic drugs, and the interaction between these drugs and Intralipid^®^ was explored. This experimental approach allowed for valuable insights into the mechanisms of intestinal lymphatic uptake and its association with Intralipid^®^. It can be employed as a convenient in vitro physiologically based biopharmaceutical tool for preliminary assessment to anticipate the outcomes of drug delivery through the intestinal lymphatics via chylomicrons. 

## 2. Materials and Methods

### 2.1. Materials

The following chemicals were used in this study: rifampicin (≤100%, CAS: 13292-46-1) was obtained from EMD Millipore Corp., Burlington, MA, USA; quercetin (≥95%, CAS: 117-39-5), pluronic L-81 (PL-81) (CAS: 9003-11-6), cannabidiol (CAS: 13956-29-1), and 1-octanol (99%, CAS: 111-87-5) were from Sigma-Aldrich Co. (Saint Louis, MO, USA); Halofantrine (CAS: 69756-53-2) was from SmithKline Beecham Pharmaceuticals (Brentford, London, UK); Intralipid^®^ 20% was from Fresenius Kabi (Toronto, ON, Canada); and peanut oil product was purchased from a local Edmonton grocery. For HPLC analysis, water (99.9%, CAS: 7732-18-5), acetonitrile (99.9%, CAS: 75-05-8), methanol (99.9%, CAS: 67-56-1), acetic acid (≥99.7%, CAS: 64-19-7) and o-phosphoric acid (85%, CAS: 7664-38-2) were of HPLC grade from FisherFisher Chemical™ (Fisher Scientific, Ottawa, ON, Canada); all other reagents were of analytical grade. 

### 2.2. Methods

#### 2.2.1. Franz Cell for Investigating Intestinal Lymphatic Uptake

The Franz Cell receiver compartment was filled with 12 mL of either Intralipid^®^ or Intralipid^®^ mixed with a potential inhibitor or enhancer and maintained at 37.0 ± 0.5 °C with magnetic stirring at 600 rpm, as shown in Figure 2. A synthetic 0.22 µm Polyvinylidene Fluoride (PVDF) membrane impregnated with octanol was placed between compartments. For the uptake experiments four model drugs (cannabidiol, halofantrine, quercetin and rifampicin) were used and to capture the inhibition and enhancement effect, rifampicin was utilized. Two mL solutions of the model drugs were added to the donor compartment with the receiver compartment containing either Intralipid^®^ or Intralipid^®^ mixed with an enhancer or an inhibitor. At different time intervals (0–4 h), 0.2 mL samples were withdrawn, extracted, and analyzed for drug content using HPLC with a C18 column (150 mm × 4.6 mm i.d., 5 µm). The column temperature was maintained at 25 °C, and the conditions of analysis for all drugs are listed in Table 1.

#### 2.2.2. Entrapment Efficiency

To determine the entrapment efficiency of rifampicin in Intralipid^®^, a solution of rifampicin was mixed with Intralipid^®^ at a concentration of ~145 mg/mL and stirred for 15 min. Then, 0.5 mL of the mixture was added to an Amicon Ultra-0.5 30 KDa centrifugal filtering unit (Millipore, Sigma-Aldrich, Darmstadt, Germany) and centrifuged at 10,000× *g* for 10 min. The filtrate was collected and analyzed using HPLC. The sample was then returned to the filtering unit and the process was repeated. The filtrate was collected again and analyzed to determine the entrapment efficiency.

#### 2.2.3. Characterization of Intralipid^®^

##### Measurement of Size of Intralipid^®^

The particle size of Intralipid^®^ and Intralipid^®^ mixed with different percentages of pluronic L-81 (0.05%, 0.1%, 1% and 10%) were measured using a Malvern Ultra Zeta Sizer (Malvern, UK) at an angle of 173° and 25 °C. This instrument uses 10 mW 632.08 nm HeNe laser, adaptive correlation algorithm, and avalanche photo diode (APD) detector. Samples were analyzed in polystyrene latex cells (DTS0012) for 30 runs in triplicates for each. Results were analyzed using Malvern Panalytical software (version: 2.1.0.15).

##### Morphological Characterization of Intralipid^®^ via Transmission Electron Microscopy (TEM)

The morphology of various Intralipid^®^ samples was studied using transmission electron microscopy (TEM) following a negative staining procedure. A drop of the sample was placed on a 300-mesh, carbon-coated copper grid, the excess solution was removed using blotting paper, and the samples were stained with a drop of 1% phosphotungstic acid for 60 s. The stained samples were dried at ambient temperature and observed with TEM (JEM-1230, JEOL Ltd., Akishima, Tokyo, Japan) at an acceleration voltage of 120 kV.

### 2.3. Statistical Analysis

Statistical comparisons were performed using GraphPad Prism software version 10.10.3 (GraphPad Software, San Diego, CA, USA). Paired t-tests were employed for comparisons between two groups, while one-way ANOVA was used for multiple groups. A significance level of α = 0.05 was applied, and in all instances, *p*-values of less than 0.05 were considered indicative of statistical significance.

## 3. Results and Discussion

### 3.1. Lymphatic Uptake via the In Vitro Model

The chylomicron-drug association has proven to be an effective approach for investigating the intestinal lymphatic absorption of drugs [23,24]. Therefore, this in vitro model was developed to replicate this in vivo process. Since certain lipophilic drugs are taken up from the intestinal lumen into enterocytes and subsequently enclosed within chylomicrons [3], the developed model was designed to include a donor compartment, resembling the intestinal lumen, where the drug solution is placed. Additionally, there was a receiver compartment that contains an artificial chylomicron medium, simulating the environment within enterocytes. These two compartments were separated by an octanol-immersed membrane, which served as a representation of the lipophilic cell membrane found in enterocytes (Figure 2).

The results of the release of various drugs into the receiver compartment of the proposed intestinal lymphatic uptake model are shown in Figure 3 and indicate significant differences (*p* = 0.0236) among the drugs in terms of lymphatic uptake. Halofantrine showed the highest uptake, with nearly all drugs detected in the artificial chylomicrons compartment after 2 h. Quercetin and cannabidiol showed moderate efficiency in passing through the side of the artificial chylomicrons, while rifampicin was the least available. The entrapment efficiency results suggested the uptake of the drugs into the Intralipid^®^ particles, as none was detected outside.

The examined drugs were individually tested and suggested in previous studies to be lymphatically absorbed. Halofantrine has been extensively investigated for lymphatic transport, as its bioavailability was improved due to lymphatic voyage [26,27,28], and its superiority over other tested lymphotropics in the proportion traversing into the lymphatics is established [1,29]. Investigative reports on cannabidiol, quercetin, and rifampicin also documented the contribution of lymphatic uptake to their circulating plasma concentrations [30,31,32,33].

Halofantrine is a lipophilic antimalarial that has prompted several reports underscoring the impact of lymphatic transport on its overall oral bioavailability. In a specific study utilizing lymph-cannulated rats, the administration of halofantrine in lipidic vehicles resulted in 15.8% of the drug’s journey occurring through the lymphatic system, while the total systemic exposure was documented to be 22.7% of the administered dose [28]. A parallel experiment with halofantrine in a lipid-based formulation of long-chain triglycerides yielded values of 5.5% and 12.9% for direct systemic circulation transport and lymphatic transport, respectively. The total bioavailability documented in the study (19.1%) was mostly contributed to by the lymphatic voyage [34]. Generally, the absorption of halofantrine from the gastrointestinal tract tends to be inconsistent, but the presence of food can remarkably boost its absorption (by 6–10 times) [35]. The pivotal factor contributing to the better and more consistent oral absorption of halofantrine with food lies in its transit through the intestinal lymphatic system. When halofantrine takes the lymphatic route instead of entering through the portal circulation, it evades the extensive liver metabolism it typically undergoes, aligning with the outcomes reported.

Dietary fats have been found to enhance the oral bioavailability and systemic exposure of quercetin, a potent antioxidant flavonol. This effect is attributed to the increased lymphatic transport of quercetin, which allows it to bypass the initial metabolism in the liver [36,37]. In experiments involving thoracic duct-cannulated rats that received intraduodenal doses of quercetin with soybean oil, the transport of quercetin through the lymphatic system was notably improved [30]. Furthermore, other research has indicated that prior exposure to a high-fat diet enhances the effects when quercetin is administered. In a mesenteric-cannulated rat model receiving a dosage of 30 mg/kg of intraduodenal quercetin, the maximum concentration (Cmax) of quercetin found in the lymph was approximately five times higher than that in the plasma [36].

The non-psychoactive compound cannabidiol has been a subject of research for its potential therapeutic effects in recent decades. The reported bioavailability of cannabidiol has been less than 10% [38,39]. However, cannabidiol is highly lipophilic and has been shown to undergo significant transport through the intestinal lymphatic system when taken orally with long-chain triglycerides [40]. In experiments with Sprague-Dawley rats, it was found that co-administering cannabidiol with lipids, as opposed to using lipid-free formulations, increased its systemic exposure by a factor of three [39]. The use of oils containing long-chain fatty acids that are packaged into chylomicrons served as the basis for developing the FDA-approved oral solution of cannabidiol known as Epidiolex^®^. This cannabidiol oil solution is recommended as an antiepileptic medication for the treatment of Dravet syndrome in children [41]. 

Rifampicin, a lipophilic bactericidal drug commonly used to treat active mycobacterial infections, faces several challenges, including poor and unpredictable bioavailability and a short biological half-life. These issues can result in subtherapeutic drug levels in the bloodstream and an increased risk of developing multidrug-resistant tuberculosis (MDR-TB) [42,43]. Incorporating rifampicin into lipid-based nano-formulations has been explored as a solution to enhance its oral bioavailability. These nano-formulations help protect the drug from degradation in acidic pH environments, and the lipid component primarily aids in improving absorption through the lymphatic system [44]. Researchers have developed solid lipid nanoparticles (SLNs) and niosomes with the aim of enhancing the lymphatic uptake of rifampicin [31]. 

When analyzing the in vivo data of the tested drugs, direct comparisons and correlations with their in vitro counterparts can pose challenges. The research landscape reveals variations in drug concentrations, co-administered fats, animal models, and experimental setups for different drugs within the literature. Nevertheless, examining the highest reported fractions of intestinal lymphatic uptake for halofantrine, quercetin, and cannabidiol when co-administered with long-chain fatty acids or meals containing them, a notable pattern emerges. Their order of release in the in vitro model aligns with their respective order in vivo, with halofantrine showing a ten-fold increase in absorbed amount [35], quercetin demonstrating a five-fold increase in plasma concentration when factoring in lymphatic uptake [36], and a three-fold increase in systemic exposure for lipid-based CBD formulations [39]. Although direct comparisons may not hold significant meaning, a discernible trend emerges for these three drugs. Conversely, rifampicin lacks sufficient data on intestinal lymphatic uptake with food or long-chain fatty acids, the criteria on which the aforementioned comparisons are based, preventing the establishment of a correlation. 

Rifampicin ranked last when using molecular descriptors to correlate the order of the tested drugs with their expected in vivo outcomes. The structures of the different molecules tested, along with their molecular descriptors, are depicted in Table 2. As reported earlier, the degree of the effect sequence was found to be as follows: hydrogen-binding acceptors (HBA) > polar surface area (PSA) > solubility in long-chain triglycerides (SLCT) > logP > melting point (MP) > logD > molar volume (MV) > density > pKa > molar weight (MW) > freely rotatable bonds (FRB) > hydrogen binding donors (HBD) [24]. It has also been suggested that while HBA, PSA, HBD, MP, density, pKa, FRB, and HBD negatively affect drug association with chylomicrons, other descriptors increase it.

Considering the drugs examined, it can be noted that rifampicin had the highest values for a higher number of factors that can adversely affect release, namely, hydrogen bond acceptors and polar surface area, in addition to molecular weight, molecular volume, and hydrogen bond donors. Although halofantrine did not have the greatest number of proposed factors that aid in drug association with chylomicrons, it had the highest logP value, which is thought to enable it to penetrate the octanol-immersed membrane more easily than others, aiding in its penetration into the lipophilic core of the artificial chylomicron vesicles. Quercetin and cannabidiol were in between in terms of their release, and this can be justified by the values of their descriptors, which lie in-between rifampicin and halofantrine.

Given that rifampicin exhibited the slowest release, it was selected for further investigation to explore both inhibition and enhancement possibilities.

### 3.2. Inhibition of the Lymphatic Uptake in the In Vitro Model

Pluronic L-81 (PL-81) is a non-ionic surfactant consisting of 10% ethylene oxide (EO) and 90% propylene oxide (PO) copolymers arranged in a tri-block structure with the hydrophobic PO component located at the core of the two hydrophilic EO chains [45]. PL-81 has been demonstrated as an in vivo inhibitor of fat absorption and chylomicron secretion [12,46]. In the developed in vitro model, when PL-81 was introduced to the receiver compartment medium, a decrease in lymphatic uptake of rifampicin was observed. Altering concentrations of PL-81 resulted in varying degrees of inhibition, with both 1% and 10% causing complete inhibition of release, as shown in Figure 4.

PL-81 has been proposed to act in vivo to inhibit lymphotropic drug absorption by alternative mechanisms [47]. It is thought that it might disturb the stability of the surface of the triglyceride particles, induce their aggregation, and thus prevent the formation of the chylomicron [48], or it might interfere with the transport of triglycerides from the cytosol to the endoplasmic reticulum and consequently inhibit the formation of the triglyceride-based lipoproteins (chylomicrons) [49]. It has also been suggested that PL-81 may affect chylomicron formation by changing the conformation of the associated protein (apolipoprotein) [48].

The developed model suggested that the concentration of PL-81 plays a critical role in its effectiveness in blocking intestinal lymphatic uptake. In these two concentrations (10% and 1%), there was a statistically significant difference between the sample groups with the inhibitor and the ones without the inhibitor (* *p* < 0.05). While there was inhibition observed with 0.1% PL-81 (57.29 ± 6.25%), no inhibition was observed with 0.05% PL-81. The lack of statistical significance (*p* > 0.05) in both cases indicated that the inhibition observed may not be consistent or substantial enough to be considered significant in the two scenarios, respectively. 

Based on the obtained data, there emerged a suggestion that PL-81 might employ an additional, previously undocumented mechanism. This mechanism involves encapsulating Intralipid^®^ particles, thereby preventing the drug from entering the chylomicrons. Pluronic L-81 is postulated to coat the Intralipid^®^ droplets, as illustrated in Figure 5. The particle size results demonstrated that without PL-81, the Intralipid^®^ droplets had discrete edges and smaller sizes. But with the addition of 1% and 10% pluronic, the particle size increased from 477.86 ± 5.19 nm to 646.9 ± 4.87 nm and 1615 ± 8.1 nm, respectively (*p* = 0.0424). TEM images in Figure 6 revealed that particles with PL-81 are larger and surrounded by a layer that makes their edges less apparent, with agglomerations of particles enveloped by the same layer also noted.

### 3.3. Enhancement of the Lymphatic Uptake in the In Vitro Model

It has been reported that food, especially a high-fat diet, may affect the intestinal lymphatic uptake of drugs [37,50]. The mechanism may involve the increased formation of chylomicrons following a high-fat intake, particularly those providing the appropriate fatty acids or triglycerides to produce chylomicrons [30,51]. The length of fatty acid chains has been determined to correlate with the efficiency of lymphatic drug transport, with longer fatty acids resulting in higher lymphatic uptake. This is attributed to the increased lipophilicity of longer fatty acids, which have a higher affinity to form chylomicrons [28].

Vegetable oils, being rich in triglycerides, have been shown to stimulate chylomicron formation and enhance the oral absorption of some lipophilic drugs suggested to be absorbed via the intestinal lymphatics. For example, a three-fold improvement in the bioavailability of a synthetic cannabinoid (PRS-211,220) was achieved by administering it in a peanut oil solution [52]. Sesame oil was also documented to induce lymphatic transport of cannabidiol, resulting in a 250-fold increase in its plasma concentration and a 2.8-fold increase in systemic exposure [39,40].

When peanut oil was added to the Intralipid^®^ in the receiver compartment, the lymphatic uptake of the tested drug (rifampicin) through this model revealed a considerable improvement (*p* = 0.0276) as seen in Figure 7. Peanut oil was chosen for two main reasons: it is well-documented to facilitate intestinal lymphatic uptake, and it is composed of a variety of long-chain fatty acids (as seen in Figure 8) that resemble those found in soybean oil, which help form the artificial chylomicrons (Intralipid^®^) [53,54]. Soybean oil contains five main fatty acids, including palmitic acid (10%), stearic acid (4%), oleic acid (18%), linoleic acid (55%), and linolenic acid (13%) [55]. The fatty acids with the highest percentages are similar in both peanut and soybean oils, which are linoleic and oleic acids [56]. 

With the addition of oil, rifampicin was emulsified into the oil droplets of the Intralipid^®^, creating a rifampicin-carrier area in the receiver compartment, which eventually led to higher rifampicin movement into the receiver compartment. Figure 9 shows the difference in droplet contrast at both magnification powers; with the added oil, the center of the droplets appears clearer and shinier, indicating that the oil was emulsified into the center of the Intralipid^®^ droplets. In contrast, without added oil, the centers of the droplets appear rather dull.

While it is important to acknowledge that lipid digestion and incorporation into chylomicrons for absorption are more intricate processes in vivo, as discussed earlier, this approach can still serve as a useful tool for examination. The inclusion of lipolysis would enhance this model and provide a more comprehensive understanding of the effects of oils in vitro.

The in vitro model developed in this study, is utilizing a commercial product (Intralipid^®^), has the potential to become an invaluable tool for pharmaceutical researchers. It offers a means to investigate lymphatic drug transport—a field where conducting in vivo studies in preclinical animal models or clinical trials is challenging. This model represents the initial step towards creating a physiologically based predictive tool, which could be further refined to yield more accurate estimations of drug interaction with chylomicrons and subsequent lymphatic transport. Notably, the current in vitro model does not include a compartment for drug metabolism. This omission warrants discussion, and there is an opportunity to potentially adapt the model by incorporating elements like microsomes to better mirror in vivo conditions. Another adaptation could also encompass lipolysis, which could account for the major intricacies of the lipid gastrointestinal pathway.

## 4. Conclusions

The developed in vitro model appears to be a valuable tool for investigating and predicting lymphatic drug uptake via chylomicrons and assessing the impact of various excipients on this process. The tested drugs demonstrated different degrees of lymphotropic affinity, which highlights the importance of considering this pathway when designing drug delivery systems. The use of the lymphatic uptake inhibitor pluronic L-81 demonstrated its potential to inhibit drug uptake via the lymphatic system by coating the Intralipid^®^ particles, a finding not previously reported. The addition of peanut oil to the Intralipid^®^ in the receiver compartment demonstrated an improvement in lymphatic uptake, suggesting that the model can be utilized to explore the impact of solubility enhancers, novel formulations, and food effects, particularly fat-containing meals that may stimulate lymphatic uptake. Future studies may include exploring the effect of other excipients and formulations on lymphatic uptake as well as investigating the impact of food on lymphatic drug absorption. The model can also be utilized to screen potential lymphatic targeting agents and assess their effectiveness in improving lymphatic drug uptake. In summary, this in vitro model provides a promising platform for evaluating the lymphatic uptake of drugs and their potential for targeted drug delivery.

## Figures and Tables

**Figure 1 pharmaceutics-15-02532-f001:**
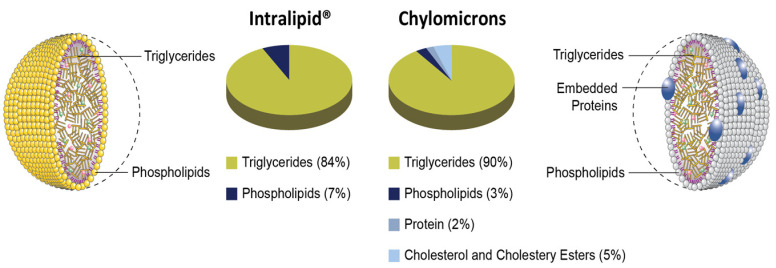
Schematic representation of natural chylomicrons (**right**) and the artificial counterpart (**left**) demonstrating their resemblance. The schematic representation of the chylomicron on the (**right**) is available under a cc license at http://cnx.org/content/col11496/1.6 (accessed on 17 September 2023). Intralipid^®^ representation on the (**left**) was created based on the existing data regarding the structure of Intralipid [25].

**Figure 2 pharmaceutics-15-02532-f002:**
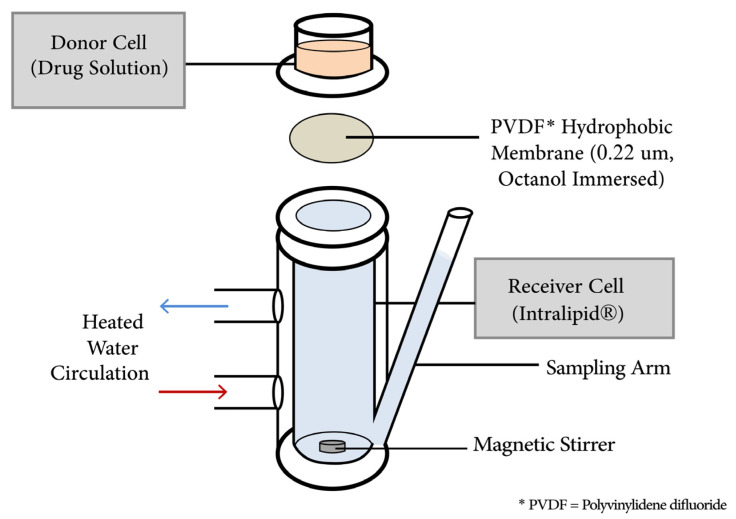
Demonstration of the proposed lymphatic uptake model.

**Figure 3 pharmaceutics-15-02532-f003:**
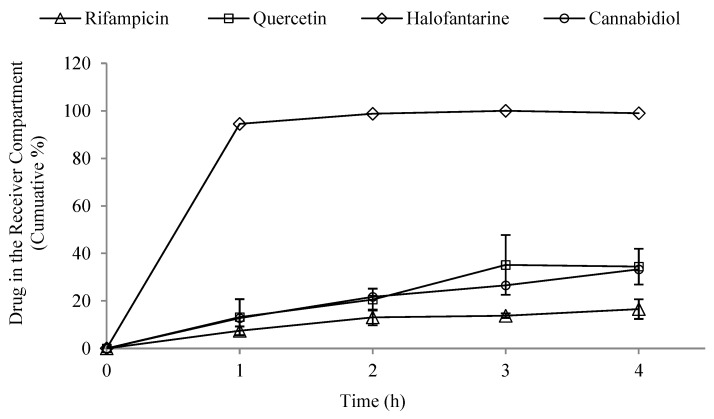
Cumulative percentage of tested drugs (cannabidiol, halofantrine, quercetin, and rifampicin) in the receiver compartment of the developed in vitro lymphatic uptake model. Data represent mean ± standard error (SE) (*n* = 3 for all drugs and 2 for halofantrine). Results showed significant differences (*p* < 0.05) among the drugs regarding their lymphatic uptake.

**Figure 4 pharmaceutics-15-02532-f004:**
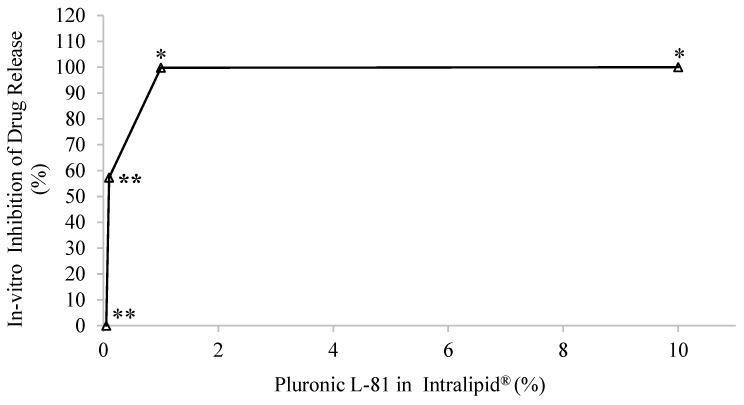
The effect of different percentages of pluronic L-81 added to Intralipid^®^ on drug release inhibition in the in vitro lymphatic uptake model. Higher concentrations (10% and 1%) were highly effective in imparting the inhibition (* *p* < 0.05). Lower concentration of the pluronic L-81 (0.1%) showed inhibition but did not reach statistical significance (** *p* > 0.05). No inhibition was seen with the 0.05% of pluronic L-81 as confirmed through statistical analysis (** *p* > 0.05).

**Figure 5 pharmaceutics-15-02532-f005:**
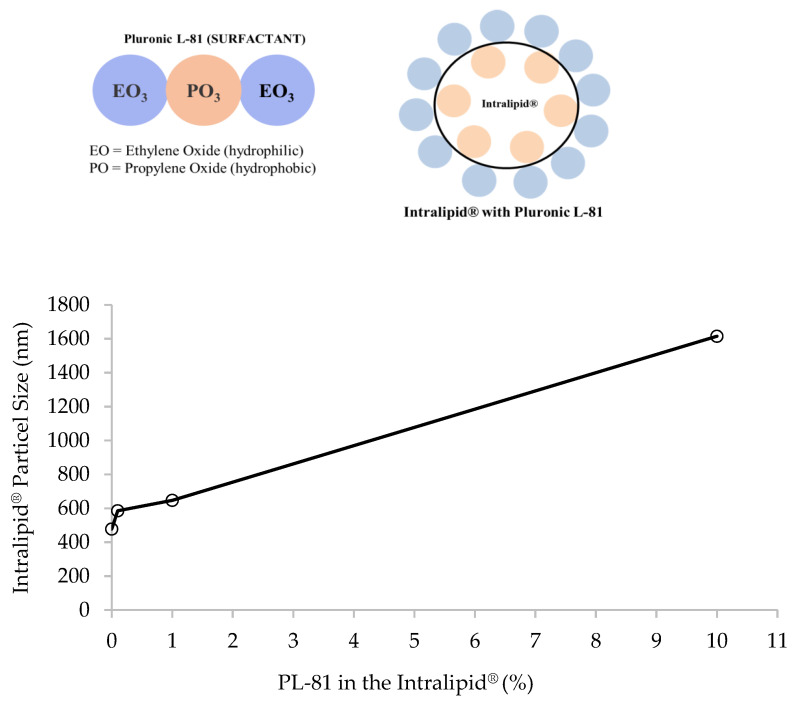
Change in the Intralipid^®^ particle size with the varying percentages of Pluronic L-81 (*p* < 0.05). As the percentage of Pluronic L-81 increased, the particle size also increased. Specifically, the presence of 0%, 0.1%, 1%, and 10% pluronic L-81 within the Intralipid^®^ yielded particle sizes of 477.86 ± 5.19 nm, 585.7 ± 5.06 nm, 646.9 ± 4.87 nm, and 1615 ± 8.1 nm, respectively (mean ± SE), *n* = 3). It was postulated that pluronic L-81 coats Intralipid^®^ particles as shown in the illustration on top.

**Figure 6 pharmaceutics-15-02532-f006:**
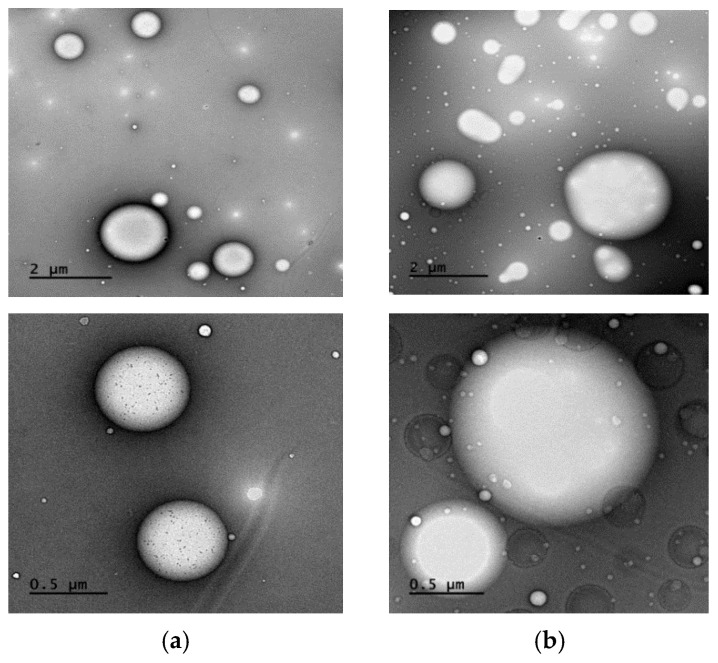
TEM images of Intralipid^®^ particles (**a**) and Intralipid^®^ with 10% pluronic-81 (**b**) at 10 K magnification power (**up**) and 40 K magnification power (**down**). The Intralipid^®^ droplets when captured alone (**a**) had discrete edges and smaller sizes, while in the presence of pluronic-81 (**b**), particles were larger and surrounded by a layer that made their edges less apparent. Agglomerations of particles enveloped by the same layer can also be noted in the presence of pluronic-81.

**Figure 7 pharmaceutics-15-02532-f007:**
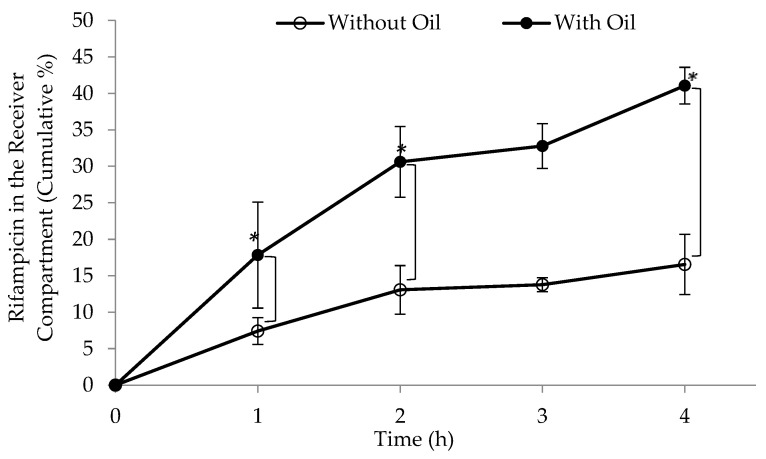
Demonstration of the increased in vitro lymphatic uptake of rifampicin via the developed model when peanut oil (2%) was added to Intralipid^®^ (*p* < 0.05). Data represent mean ± SE (*n* = 3). Data at 1, 3 and 4 h demonstrated statistical significance (* *p* < 0.05) between the samples with and without the oil.

**Figure 8 pharmaceutics-15-02532-f008:**
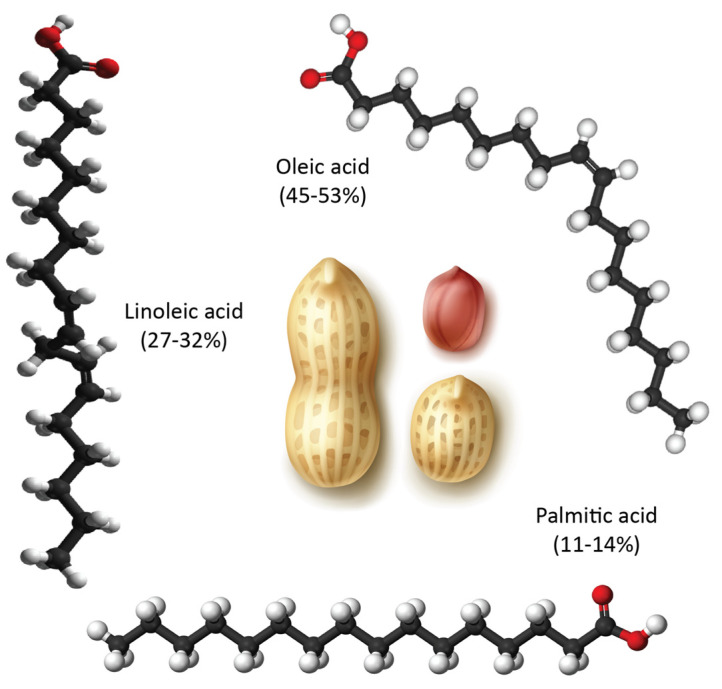
Illustration of the main fatty acids in peanut oil. This oil also contains other long-chain saturated fatty acids in small percentages such as arachidic acid (1–2%), behenic acid (1.5–4.5%), and lignoceric acid (0.5–2.5%). Credit for the peanut image is attributed to macrovector on Freepik.

**Figure 9 pharmaceutics-15-02532-f009:**
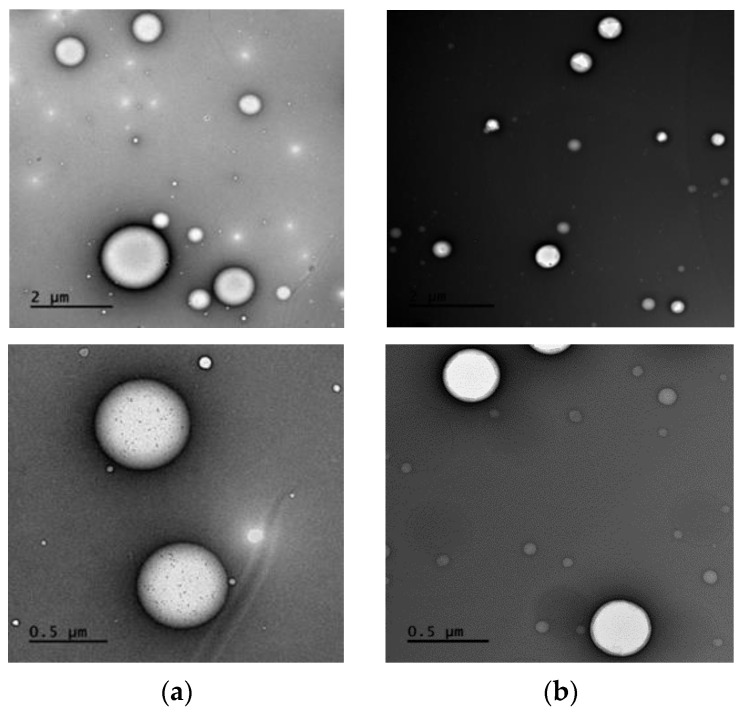
TEM images showing Intralipid^®^ particles (**a**) and Intralipid^®^ with 2% peanut oil (**b**) at 10 K magnification power (**up**) and 40 K magnification power (**down**). With the incorporation of oil, the centers of the Intralipid^®^ droplets exhibit a distinct clarity and shine, indicating emulsification of the oil into the droplet cores. Conversely, in the absence of added oil, the centers of the droplets appear dull.

**Table 1 pharmaceutics-15-02532-t001:** HPLC Analysis Conditions for the Model Lymphotropic Drugs.

Model Drug	Mobile Phase	Flow Rate (mL/min)	Detection Wavelength (nm)
Rifampicin	Methanol, Acetate Buffer (pH = 5.8) (60:40)	1.2	254
Quercetin	Methanol, Acetate Buffer (pH = 5.8) (60:40)	1.2	257, 370
Cannabidiol	Acetonitrile, Phosphoric acid (0.2%) (72:28)	1	210, 224
Halofantrine	Methanol, Phosphate Buffer (pH = 7.5) (80:20)	1	210, 259

**Table 2 pharmaceutics-15-02532-t002:** Molecular Descriptors of Tested Lymphotropic Drugs.

Drug	MW	HBA	PSA	LogP	MP (°C)	Density (g/cm^3^)	pKa	HBD	Structure
Rifampicin	822.9	15	220.15 *	4.9	183	1.178 **	1.7 7.9	6	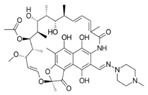
Cannabidiol	314.469 *	2 *	40.46 *	6.3 *	66–67 ^#^	1.04 ^#^	9.13 *	2 *	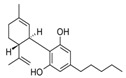
Quercetin	302.23	7	127.45 *	1.48	316–318	1.8 ^##^	7.17 8.26 10.13 12.30 13.11	5	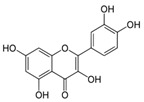
Halofantrine	500.4	5	23.5 *	8.9	93–96 and 203–204 (for the hydrochloride salt) **	1.2 ***	10.05 * 14.47	1	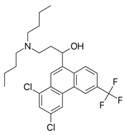

Abbreviations: molecular weight (MW), hydrogen-binding acceptors (HBA), polar surface area (PSA), melting point (MP), and hydrogen binding donors (HBD). Data was obtained from PubChem, if certain data was not found there then DrugBank *, ChemBK **, ChemSpider databases ***. The following sources were also consulted: ^#^
https://www.drugfuture.com/chemdata/Cannabidiol.html (last accessed on 12 September 2023). ^##^
https://www.chemsrc.com/en/cas/117-39-5_947030.html (last accessed on 12 September 2023).

## Data Availability

Not applicable.

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
