# Peer review of "Development of a Novel In Vitro Model to Study Lymphatic Uptake of Drugs via Artificial Chylomicrons"

_pharmaceutics, 2023, doi:10.3390/pharmaceutics15112532_

Round 1

Reviewer 1 Report

In this manuscript, the authors performed comprehensive study on the development of a model to study lymphatic uptake of drugs. Suggestions are below.

The abstract should be shortened for the maximum limit of 200 words.

The significance of cannabidiol, halofantrine, quercetin, and rifampicin - why these and not other drugs?

Introduction is too general and it not contains any data about the studied drugs (cannabidiol, halofantrine, quercetin, and rifampicin). The last paragraph of this section is usually used to present the aim of the research.

Section 2.1 Materials does not present any data about the purity and concentration of raw materials, about any previous purification, about the auxiliary compounds (octanol, the enhancer and inhibitor of Franz Cell, substances from mobile phase in HPLC etc.)

Section 2.2.2. Entrapment Efficiency Centrifugation at x10,000 g is proper for this purpose? Please present other studies that used the same or similar procedure

Section 2.2.3.1 Size Measurements

In size measurements many other parameters are important to be mentioned (time interval, channels number, laser power, acquisition mode and analysis mode / algorithm)

Section 2.2.3.2 TEM

Please indicate the same data for all instruments. Example: who is the manufacturer of JEM-1230?

Section 3.1 in-vitro Model

Figure 3 does not contains any statistics; are the differences significant or not?

The vertical bars are not explained (standard deviations or standard errors)

Section 3.2 Inhibition

Figure 4 Please use the measure units on the vertical axis and not on the curve

On the other axis there is no measure unit

Figure 5 the same situation on the horizontal axis

Section 3.3 Enhancement of the Lymphatic Uptake

Figure 7. Use stars or other symbols to present statistics

Referrences

You must pay attention to how old the quotes are (just 25% are from the last 5 years)

Reviewer 2 Report

The work conducted by Yousef et al. reported the development of in vitro model to investigate the uptake of the hydrophobic drugs into the lymphatic system. The work is well-organized and would benefit the scientific community. However, the following concerns should be critically reviewed and addressed before this work can be published.

1.     Since LgP and TG solubility are the most frequently reported key factors for predicting drug lymphatic transport, the authors should detect/state the LgP and triglycerides solubility of the investigated drugs. Do the investigated drugs meet the solubility standards?

2.     Full Dissolution study should be conducted for all investigated drugs in the designed system should be conducted. (Higushi model or other), to measure the ability to maintain the drug in a solubilized state during in-vivo lipolysis devoid of precipitation.

3.     An in vitro biological application should be involved to prove the hypothesized model. The improved lymphatic uptake should be tested on different cells (https://doi.org/10.1007/s11095-014-1578-x).

4.     The authors should conduct a release kinetics study involving at least 4 models. The release kinetics will definitely give more insights into the proposed system.

5.     The drug precipitation kinetics should be studied using Raman spectroscopy and/or Fluorescence resonance energy transfer.

6.     The in vitro studies should be correlated to in vivo studies, such as measuring the pharmacokinetic parameters viz. Cmax, Tmax, and AUC,

7.     The authors should involve an ex-vivo model to this study, such as a gut sac model. Ex vivo models are able to correlate the in-vivo condition more realistically and they guide for drug absorption at the organ level. These models are advantageous when the drug under investigation is hydrophobic, which is the case.

8.     The authors should add a statistical analysis section in the methodology to describe the adopted statistical model and probability.

9.     The whole study lacks proper statistical analysis, whereas most figures lack the standard error bars. Also, the authors should assess statistical significance in all figures, and at what P value.

10.  The authors should perform all experiments at least thrice (three independent experiments) and plot the quantitative values.

11.  Please adjust the y-axis label for Figures 3, 5 and 7.

Minor edits

Round 2

Reviewer 1 Report

Dear authors,

I congratulate you for the changes made and I believe that the value of the manuscript has increased in this way

Reviewer 2 Report

The authors responded to most of the suggestions to improve the paper. I recommend acceptance in the current form.

Minor edits